# CLEC12A Binds to *Legionella pneumophila* but Has No Impact on the Host’s Antibacterial Response

**DOI:** 10.3390/ijms24043891

**Published:** 2023-02-15

**Authors:** Ann-Brit Klatt, Christina Diersing, Juliane Lippmann, Sabine Mayer-Lambertz, Felix Stegmann, Swantje Fischer, Sandra Caesar, Facundo Fiocca Vernengo, Katja Hönzke, Andreas C. Hocke, Jürgen Ruland, Martin Witzenrath, Bernd Lepenies, Bastian Opitz

**Affiliations:** 1Department of Infectious Diseases, Respiratory Medicine and Critical Care, Charité—Universitätsmedizin Berlin, Corporate Member of Freie Universität Berlin and Humboldt-Universität zu Berlin, 13353 Berlin, Germany; 2Institute for Immunology, University of Veterinary Medicine Hannover, 30559 Hannover, Germany; 3Research Center for Emerging Infections and Zoonoses, University of Veterinary Medicine Hannover, 30559 Hannover, Germany; 4Max Planck Institute for Infection Biology, Vector Biology, 10117 Berlin, Germany; 5Faculty of Health Sciences Brandenburg, Brandenburg University of Technology Cottbus—Senftenberg, 03046 Cottbus, Germany; 6German Center for Lung Research (DZL), Augustenburger Platz 1, 13353 Berlin, Germany; 7Institute of Clinical Chemistry and Pathobiochemistry, School of Medicine, Technical University of Munich, 80333 Munich, Germany; 8Center for Translational Cancer Research (TranslaTUM), 81675 Munich, Germany; 9German Cancer Consortium (DKTK), Partner Site Munich, 80336 Munich, Germany; 10German Research Center (DKFZ), 69120 Heidelberg, Germany; 11German Center for Infection Research (DZIF), Partner Site Munich, 17493 Greifswald, Germany

**Keywords:** *Legionella pneumophila*, C-type lectin receptors, CLEC12A, macrophages

## Abstract

*Legionella pneumophila* is an intracellular pathogen that can cause severe pneumonia after the inhalation of contaminated aerosols and replication in alveolar macrophages. Several pattern recognition receptors (PRRs) have been identified that contribute to the recognition of *L. pneumophila* by the innate immune system. However, the function of the C-type lectin receptors (CLRs), which are mainly expressed by macrophages and other myeloid cells, remains largely unexplored. Here, we used a library of CLR-Fc fusion proteins to search for CLRs that can bind the bacterium and identified the specific binding of CLEC12A to *L. pneumophila*. Subsequent infection experiments in human and murine macrophages, however, did not provide evidence for a substantial role of CLEC12A in controlling innate immune responses to the bacterium. Consistently, antibacterial and inflammatory responses to *Legionella* lung infection were not significantly influenced by CLEC12A deficiency. Collectively, CLEC12A is able to bind to *L. pneumophila*-derived ligands but does not appear to play a major role in the innate defense against *L. pneumophila*.

## 1. Introduction

*L. pneumophila* is an intracellular bacterial pathogen that persists in the environment as a parasite of freshwater protozoans such as *Acanthamoeba castellanii* [1]. After the inhalation of contaminated aerosols from, e.g., cooling towers, hot and cold water systems and whirlpools, humans can develop an infection [2]. The severity of *Legionella* infections range from mild flu-like Pontiac fever to Legionnaires’ disease, an atypical, often severe form of pneumonia that is associated with 10% mortality [3]. Within the lung, *L. pneumophila* is engulfed by alveolar macrophages, where it replicates inside an endoplasmic reticulum (ER)-like organelle called the *Legionella*-containing vacuole (LCV) [4,5]. The establishment of the LCV and other manipulations of the host cell require the bacterial *dot/icm*-encoded type IV secretion system (T4SS), which injects around 300 effector molecules into the host’s cytosol [6,7].

The innate immune system uses various PRRs to detect *L. pneumophila* infection. These receptors include the transmembrane Toll-like receptors (TLRs) and the cytosolic NOD-like receptors (NLRs), as well as cyclic GMP-AMP synthase (cGAS), which senses, e.g., bacterial cell wall components, flagellin and nucleic acids. The sensing of *Legionella* by these PRRs initiates, for example, the production of proinflammatory cytokines and IFNs, as well as the activation of so-called inflammasomes [8,9,10,11]. Antibacterial immune defense in the lung against *L. pneumophila* partly relies on TNFα- and type I and II IFN-dependent macrophage-intrinsic resistance mechanisms, as well as on an inflammasome-mediated type of cell death named pyroptosis [12,13,14,15,16,17].

CLRs are PRRs that often bind to carbohydrate ligands, although proteins and lipids have also been identified as CLR ligands [18,19,20]. C-type lectin domain family 12 member A (CLEC12A/MICL) is a type II transmembrane protein that is primarily expressed on myeloid cells, including granulocytes, monocytes, macrophages and dendritic cells [21]. CLEC12A is an evolutionarily conserved inhibitory CLR containing an immunoreceptor tyrosine-based inhibitory motif (ITIM), thus recruiting inhibitory phosphatases upon activation [22]. While human and murine CLEC12A are structurally and functionally similar, human CLEC12A is a heavily glycosylated monomer, whereas murine CLEC12A is expressed as a less glycosylated dimer [23]. In previous studies, CLEC12A was described as a receptor involved in the regulation of homeostasis and in the control of inflammation, including rheumatoid arthritis and experimental autoimmune encephalomyelitis (EAE) [24,25]. CLEC12A was identified as a receptor for dead cells sensing uric acid crystals (monosodium urate, MSU) as a key danger signal for cell-death-induced immunity [26]. Lately, we identified CLEC12A as a receptor for another crystalline ligand, hemozoin, and showed the crucial role of CLEC12A in the development of cerebral malaria [27]. To date, there are only few studies on the role of CLEC12A in the context of bacterial infections. In a murine *Salmonella* infection model, it was shown that CLEC12A contributed to antibacterial autophagy, as CLEC12A was recruited to bacteria–autophagosome complexes and interacted with an E3-ubiquitin ligase complex functionally involved in autophagy [28]. Recently, mycolic acids from various mycobacterial species were described as binding to mouse CLEC12A and, more potently, to human CLEC12A. Innate immune responses were augmented in CLEC12A-deficient mice after *M. tuberculosis* infection, suggesting that mycobacteria dampened the host’s immune response by hijacking CLEC12A through their mycolic acids [29]. Currently, the role of CLEC12A in the recognition of *L. pneumophila* and in the induction of innate immune responses during *Legionella* infection in vivo is unknown.

Since the role of CLRs in *Legionella* infection was unknown, here, we screened for CLRs capable of binding the bacterium, and functionally characterized the role of CLEC12A in *L. pneumophila* infection. We used a comprehensive CLR-Fc fusion protein library to screen for CLRs sensing *L. pneumophila* and identified CLEC12A as a candidate receptor binding to different *L. pneumophila* strains. A distinct *Legionella*-derived ligand for CLEC12A remains to be identified; nevertheless, we could already exclude lipopolysaccharide (LPS) from *L. pneumophila* and putative (glyco-)proteins as CLEC12A ligands. In a murine *L. pneumophila* infection model, bacterial loads and effector functions such as cytokine secretion were assessed but showed no significant difference between WT and *Clec12a*^−/−^ mice. We therefore conclude that CLEC12A recognizes *L. pneumophila* but has a limited role in antibacterial defense.

## 2. Results

### 2.1. The CLR CLEC12A Recognizes L. pneumophila

To investigate whether CLRs bind to *L. pneumophila*, we initially performed a flow cytometry-based binding assay using a comprehensive CLR-Fc library [30,31]. To this end, *L. pneumophila* JR32 was incubated with the respective CLR-Fc fusion proteins. Indeed, several CLR-Fc fusion proteins (CLEC12A, Dectin-1, DC-SIGN and L-SIGN) exhibited moderate to substantial binding to *L. pneumophila*, with CLEC12A displaying the highest MFI signal compared with the hFc control and the other CLRs included in the library (Figure 1A). Since *L. pneumophila* serogroup 1 causes most of the worldwide outbreaks [32], we decided to test the wild-type strains JR32 and Corby, both categorized as serotype 1, for binding to CLEC12A [12,33]. To gain further insights into the role of major *L. pneumophila* pathogenicity factors in CLEC12A binding, a LPS mutant [33] and a flagellin mutant (*ΔflaA*) [34] were included in the binding assay. The LPS mutant TF 3/1 carries a point mutation in the *lag-1* gene, resulting in a loss of the 8-*O*-acetyl groups in legionaminic acid [33], while the flagellin mutant is characterized by an in-frame deletion of the gene *lpg1340*, leading to a cessation of flagellin expression [34]. To exclude the potential impact of the CLR-Fc’s structure on the CLR–*L. pneumophila* interaction, we used C- and N-terminal hFc-fused murine CLEC12A in the binding study. An ELISA-based binding study revealed the binding of both CLEC12A fusion protein constructs to all tested heat-killed *L. pneumophila* strains (Figure 1B). Although the TF 3/1 mutant exhibited slightly reduced binding to CLEC12A, this finding suggests that CLEC12A recognition is independent of the *Legionella* LPS. A flow cytometry-based binding study demonstrated the recognition of intact *L. pneumophila* by murine CLEC12A (Figure 1C,D), thereby confirming and extending the results obtained by the ELISA-based assay (Figure 1B). In conclusion, CLEC12A recognizes *L. pneumophila* as well as the LPS- and flagellin-deficient mutants.

To unravel the nature of the putative CLEC12A ligand in *L. pneumophila*, we extracted LPS from the wild-type strain JR32, as described by Lück et al. [35]. This extraction method allowed the separation of the *L. pneumophila* LPS into three fractions: cell-wall-bound compounds, outer membrane vesicles and soluble non-vesicular LPS. For the ELISA-based binding study, we used *E. coli* LPS as a control and compared the binding of murine CLEC12A and hFc to the different LPS preparations. As no significant binding of murine CLEC12A to LPS was observed (Figure 1E), LPS does not seem to play a major role in the interaction of *L. pneumophila* with murine CLEC12A. To test the proteinaceous nature of the putative murine CLEC12A ligand, *L. pneumophila* JR32 was lysed by sonication, followed by proteinase K digestion. The lysates were pipetted on a nitrocellulose membrane and incubated with hFc or murine CLEC12A-hFc followed by detection using an HRP-conjugated anti-hFc antibody (Figure 1F). Interestingly, CLEC12A still bound to the proteinase K-digested *L. pneumophila* lysate, thus rendering a *L. pneumophila*-derived protein ligand for CLEC12A unlikely (Figure 1F). While the exact nature of the CLEC12A ligand in *L. pneumophila* still needs to be identified, here, we identified murine CLEC12A as a CLR recognizing *L. pneumophila*.

### 2.2. CLEC12A Does Not Affect the Replication of L. pneumophila in Murine Macrophages and Infection-Induced Cytokine Responses

Given that murine CLEC12A binds *L. pneumophila*, we tested whether bacterial replication in murine macrophages or the production of cytokines was affected. Since the growth of *L. pneumophila* is restricted in C57Bl/6 macrophages by the NAIP5/NLRC4 inflammasome [15,16], *L. pneumophila ΔflaA* was used for the replication assay. The infection of bone marrow-derived macrophages (BMDMs) from WT and *Clec12a*^−/−^ mice with *L. pneumophila ΔflaA* for over 72 h did not reveal any effect of CLEC12A on bacterial replication (Figure 2A). Moreover, the *L. pneumophila*-induced expression of *Ifnb1* and the production of TNFα was not significantly influenced by a CLEC12A deficiency (Figure 2B,C). We therefore concluded that CLEC12A does not play a major role in controlling bacterial replication or in regulating cytokine production during *L. pneumophila* infection of murine macrophages.

### 2.3. The Limited Role of Human CLEC12A in L. pneumophila Infection

To investigate the role of CLEC12A during *L. pneumophila* infection in human cells, we used BLaER1-derived macrophage-like cells (Figure 3A). BlaER1-derived macrophages were generated from the immortalized B cell line BlaER1 by heterologous inducible expression of the myeloid transcription factor C/EBPα [36]. After transdifferentiation, BlaER1 macrophages are adherent, non-proliferative, highly phagocytic, have a macrophage-like transcriptional profile and morphology, and are well suited for studies of the innate immune response, as they behave more like primary human monocytes/macrophages than many of commonly used cell lines such as THP1 and U937 [37,38]. BLaER1-derived macrophages also support the replication of *L. pneumophila* to a similar extent to human alveolar macrophages and express CLEC12A (Appendix A). We constructed a BLaER1 *CLEC12A*^−/−^ line by using CRISPR/Cas9 (Figure 3B), then confirmed the deficiency in CLEC12A by flow cytometry (Figure 3C) and used the infected WT and *CLEC12A*^−/−^ cells to assess bacterial replication and cytokine production. We observed no differences in bacterial load in the WT and *CLEC12A*^−/−^ cells infected with *L.p. ΔflaA* or *ΔdotA* over the course of 72 h (Figure 3D). The *L. pneumophila*-induced expression of *IFNB1*, *IL1B* and *TNFA* (Figure 3E,G,I), and the production of the IFN-inducible cytokine IP-10 as well as IL-1β and TNFα (Figure 3F,H,J) were not significantly affected by the CLEC12A deficiency. Thus, our results do not support an important role of human CLEC12A in *L. pneumophila* infection.

### 2.4. Role of CLEC12A in Pulmonary L. pneumophila Infection In Vivo

Finally, we investigated the role of CLEC12A during *L. pneumophila* infection in vivo [9,12,17]. C57BL/6 WT and *Clec12a*^−/−^ mice were intranasally infected, and the bacterial loads in the lung were evaluated at different time points after infection. We observed a trend towards lower bacterial loads in the lungs of *Clec12a*^−/−^ mice at 48 and 96 h after infection, which, however, was not significant (Figure 4A). Monitoring of temperature and body weight did also not reveal differences between WT and *Clec12a*^−/−^ animals (Figure 4B,C). Next, we measured the gene expression and proinflammatory cytokine levels in the lungs of WT and *Clec12a*^−/−^ mice at 24 h after infection. We found that neither the expression of pulmonary *Ifnb1* nor the production of IFNy, TNFα or IL-6 were significantly influenced by a deficiency in CLEC12A (Figure 4D–G). Moreover, the ratio and number of alveolar macrophages (AMs), inflammatory monocytes (iMonos) and polymorphonuclear neutrophils (PMNs) in the lungs of WT and *Clec12a*^−/−^ mice were similar (Figure 4H–N). Taken together, our data indicate that CLEC12A does not play an important role in pulmonary *L. pneumophila* infection in mice.

## 3. Discussion

The role of CLRs in the innate immune responses to bacterial pathogens in general and to *L. pneumophila* in particular is poorly understood. We therefore screened for CLRs capable of binding to *L. pneumophila* by using a library of CLR fusion proteins and identified CLEC12A as the CLR with the strongest bacterial binding capacity. Subsequent functional experiments, however, did not reveal any influence of CLEC12A on the infection of human and murine macrophages by *L. pneumophila* or antibacterial innate immunity. Moreover, CLEC12A did not influence *L. pneumophila*-induced pneumonia in mice. We thus conclude that CLEC12A is able to bind *L. pneumophila* but does not play a role in the host’s antibacterial responses to the infection.

CLEC12A has been described as an innate immune receptor for uric acid crystals, plasmodial hemozoin and mycobacterial mycolic acids with either anti-inflammatory or adaptive immunity-promoting functions [26,27,29]. Moreover, CLEC12A was found to potentiate the type I IFN responses induced by cytosolic RNA sensors, and to stimulate antibacterial autophagy during *Salmonella* infection by interacting with the endogenous host proteins [28,39]. Considering that all of these downstream effects are also of potential importance for *L. pneumophila* infection, that CLEC12A is highly expressed in the macrophages and that it strongly binds to the bacterium, we assumed that a deficiency in CLEC12A might influence *L. pneumophila* infection and the antibacterial defense. Unexpectedly, however, we observed no effect of CLEC12A on the replication of *L. pneumophila* in both human and murine macrophages, or on bacteria-induced cytokine production. Consistent with these findings, CLEC12A deficiency did not significantly influence *L. pneumophila* lung infections in vivo, although a non-significant trend towards lower bacterial loads in *Clec12a*^−/−^ mice compared with the WT control mice was observed. Thus, while the possibility of a partly redundant immune regulatory function of CLEC12A during *L. pneumophila* lung infections cannot be fully excluded, our results do not support the important role of CLEC12A in *L. pneumophila* infection. Since *L. pneumophila* possesses RavZ-dependent and -independent mechanisms to prevent autophagic degradation of intravacuolar bacteria [40,41], it is possible that effects of CLEC12A on autophagy are masked by these bacterial mechanisms in the context of *Legionella* infection. Moreover, the role of individual CLRs in bacterial infections might also depend on the mouse model and the bacterial strain used, as recently shown, for example, for Mincle in pneumococcal infections [42,43]. Thus, we cannot exclude the possibility that we would have seen the effects of CLEC12A if we had used a different *L. pneumophila* strain.

To identify the interactions of *L. pneumophila* with the host CLRs, we used a comprehensive CLR-Fc fusion protein library that was established to identify novel CLR–ligand interactions [22,42]. Screening with this library revealed the substantial binding of CLEC12A to both intact *L. pneumophila* in a flow-cytometry based binding assay and heat-killed *L. pneumophila* in an ELISA-based study. While CLEC12A has been regarded so far as a CLR recognizing dead cells [26], pathogen-derived molecules were also reported as ligands for CLEC12A more recently [27,29]. In particular, the latter study highlighted that the recognition of CLEC12A is not restricted to ligands of crystalline nature such as uric acid or plasmodial hemozoin crystals, but it may also bind to lipidic ligands, thereby affecting the innate immune response and serving as a target for bacterial immune evasion mechanisms [29]. We initially considered *L. pneumophila* LPS as a candidate CLEC12A ligand, as it is the main antigen recognized by the antibodies contained in the serum of patients and plays a critical role in the early stages of infection by anchoring the bacteria to the host cell membrane [44]. *L. pneumophila* LPS differs from LPS from other Gram-negative bacteria by its high hydrophobicity caused by the presence of deoxy groups and N- and O-acyl substituents in legionaminic acid [44]. However, the binding of CLEC12A to the *L. pneumophila* LPS mutant TF 3/1 exhibiting a loss of the 8-*O*-acetyl groups in the legionaminic acid was only marginally affected compared with the wild-type strains Corby and JR32. Furthermore, we purified the *L. pneumophila* LPS according to established protocols and did not observe significant binding to CLEC12A in an ELISA-based assay, thus rendering a CLEC12A–LPS interaction unlikely. Subsequent binding assays evaluated by flow cytometry using a flagellin mutant (*ΔflaA*) ruled out flagellin as a CLEC12A ligand and, importantly, a dot-blot-based lectin assay using proteinase K-treated *L. pneumophila* lysates even excluded (glyco-)proteins as putative CLEC12A ligands. In light of a recent publication showing the binding of CLEC12A to mycolic acids from various mycobacterial species [29], it would be interesting to isolate *L. pneumophila*-derived lipid fractions and test them for binding to CLEC12A and for agonistic activity. However, a distinct CLEC12A ligand identification would go beyond the scope of the present study.

In order to examine CLEC12A’s functional role in human macrophages, we employed CRISPR/Cas9 technology to generate human CLEC12A-deficient BLaER1-derrived macrophages. BlaER1 macrophages have emerged as a suitable model for the study of innate immune recognition, since they behave more similarly to primary human monocytes/macrophages than many of commonly used cell lines such as THP1 and U937, and because genetic loss-of-function studies are possible [37,38]. They are adherent, non-proliferative and highly phagocytic, and develop a macrophage-like transcriptional profile [38]. Moreover, we demonstrated that BLaER1-derived macrophages express CLEC12A and enable the replication of *L. pneumophila*, similar to human alveolar macrophages. Thus, BLaER1-derived macrophages represent a suitable model for studying *L. pneumophila* infection in human macrophages.

In summary, our study identified that CLEC12A binds to *L. pneumophila*. While the exact *Legionella* ligands of CLEC12A remain unidentified, both LPS and (glyco-)proteins are unlikely to be involved in the binding of CLEC12A. Our functional experiments in murine and human macrophages, as well in vivo experiments in mice, did not provide any evidence for an important functional role of CLEC12A in *L. pneumophila* infection.

## 4. Materials and Methods

### 4.1. Ethical Statement

All animal experiments were carried out with strict adherence to German law (Tierschutzgesetz, TierSchG), following the approval of the corresponding institutional (Charité-Universitätsmedizin Berlin) and governmental animal welfare authorities (LAGeSo Berlin, approval ID G0334/17). Experiments with human alveolar macrophages were performed with the ethical approval of Charité EA2/079/13.

### 4.2. Bacteria and Culturing

The *L. pneumophila* strain JR32 of serogroup type I was used, as well as the isogenic mutants *ΔflaA* and *ΔdotA*. Further, the Corby strain and the isogenic mutant TF 3/1 were used [33]. The isogenic mutant TF3/1 was kindly provided by Dr. Christian Lück (Technische Universität Dresden). The *L. pneumophila* strains were cultured on buffered charcoal yeast extract (BCYE) agar plates for two days at 37 °C. For the binding studies, bacteria were grown in a medium of N-(2-acetamido)-2-aminoethanesulfonic acid (ACES)-buffered yeast extract (AYE) and were washed twice with PBS. Both the BCYE agar plates and the AYE medium were supplemented with L-cysteine and ferric nitrate. Heat-killing was performed at 75 °C for 1 h and verified by plating the bacteria on BCYE agar.

### 4.3. Mice

The generation of the *Clec12a*^−/−^ mice was described previously [26] and the mice were backcrossed to a C57BL6/J background. The genotype of the *Clec12a*^−/−^ mice was confirmed by PCR and flow cytometry analysis of spleen and bone marrow cells using PE-conjugated anti-mouse CLEC12A antibody (5D3/CLEC12A, Bio-Legend, San Diego, CA, USA, 1:200) [27]. C57BL6/J WT and *Clec12a*^−/−^ mice were bred at the Institute for Biochemistry at Stiftung Tierärztliche Hochschule Hannover. Infection experiments were performed at the Charité-Universitätsmedizin Berlin, Department of Infectious Diseases, Respiratory Medicine and Critical Care.

### 4.4. Murine Model of Legionnaires’ Disease

Anesthetized 8- to 16-week-old female WT and *Clec12a*^−/−^ mice were intranasally infected with 10^6^ CFU *L. pneumophila* JR32 suspended in 40 µL PBS. Control mice were treated with 40 µL PBS. Mice were sacrificed at 24 and 48 h after infection, and their bacterial loads were evaluated by lysing cells of a homogenized lung suspension in 0.2% Triton X-100 and then plating a defined volume on BYCE agar. Additionally, cytokine and mRNA levels as well as the numbers of different cell populations were evaluated after 24 h and compared with the PBS-treated control group.

### 4.5. CLR-Fc Fusion Proteins

The production of CLR-Fc fusion protein was performed as described [30,31]. Briefly, cDNA encoding the C-type lectin-like domain (CTLD) of the respective CLR was ligated into the pFuse-hIgG1-Fc (hFc) expression vector (Invivogen, Toulouse, France). For the murine CLEC12A (mCLEC12A)–C-terminal hFc fusion protein (N-CTLD-hFc-C), the amplified cDNA was fused to the N-terminus of hFc. Vice versa, for N-terminal hFc fusion proteins (N-hFc-CTLD-C), the cDNA was fused to the C-terminus of hFc. For protein expression, CHO cells were transfected with the respective plasmid. Proteins were purified using HiTrap Protein G HP columns (GE Healthcare, Chicago, IL, USA) [45]. The purity and identity of the fusion proteins were tested by SDS-PAGE followed by Coomassie staining and Western blotting using a HRP-conjugated goat anti-human IgG antibody (Jackson ImmunoResearch Labs, Cambridgeshire, UK, RRID: AB_2337586) [30]. Subsequently, the functionality of the fusion proteins was tested. For CLEC12A-hFc, binding studies with monosodium urate (MSU) were performed [26].

### 4.6. ELISA-Based Binding Studies

The ELISA-based binding studies were conducted as described in Mayer et al. [31]. Briefly, heat-killed bacteria or bacterial LPS from *L. pneumophila* and *E. coli* (Sigma-Aldrich, St. Louis, MO, USA) were coated on half-area microplates (Greiner Bio-One GmbH, Frickenhausen, Germany) and incubated with 250 ng of the respective fusion proteins in a lectin-binding buffer (50 mM HEPES, 5 mM magnesium chloride and 5 mM calcium chloride; pH 7.4). Binding of the fusion proteins was detected using a 1:5.000-diluted HRP-conjugated goat anti-human IgG antibody (Jackson ImmunoResearch Labs, Cambridgeshire, UK, RRID: AB_2337586). An OPD-Solution (*o*-phenylenediamine dihydrochloride substrate tablet (Thermo Fisher, Waltham, MA, USA), 24 mM oxalosuccinic acid, 0.04% H_2_O_2_ and 50 mM disodium hydrogen phosphate in H_2_O) was used as the HRP substrate, and the reaction was stopped with 2.5 M sulfuric acid. Finally, the optical density was measured at 495 nm using a Multiskan Go microplate spectrophotometer (Thermo Fisher Scientific, Waltham, MA, USA). For the LPS binding studies, substrate oxidization was measured in a kinetic loop at 450 nm.

### 4.7. Flow Cytometry-Based Binding Studies

Samples were prepared as described in Mayer et al. [31]. Live bacteria were incubated with 250 ng of fusion proteins in a lectin binding buffer. The binding of the fusion proteins was detected using a 1:200-diluted polyclonal goat anti-human IgG (Fc)-PE (Jackson ImmunoResearch Labs, Cambridgeshire, UK, RRID: AB_2337675). Samples were analyzed using an Attune NxT Flow Cytometer (Thermo Fisher Scientific, MA, USA). For data analysis, FlowJo Software (FlowJo, Ashland, OR, USA) was used.

### 4.8. LPS Extraction

Extraction of the LPS from *L. pneumophila* WT JR32 was performed according to Lück et al. [35]. Briefly, the bacterial culture was centrifuged (with the pellet considered to be cell-wall-bound components), and the supernatant was sequentially filtered through a 0.2 µm strainer and filters with a 300 and 10 kDA molecular weight cut-off (MWCO) (Corning Spin-X UF concentrators, Corning Inc., Corning, NY, USA). Fractions that were retained by the 300 kDa MWCO filter were considered to be outer membrane vesicles. Flow-through that had passed the 300 kDa MWCO filter but was retained by the 10 kDa MWCO filter was considered to be soluble, non-vesicular LPS. Both the cell-wall-bound components and the soluble non-vesicular LPS were heat-inactivated and treated with Proteinase K (Thermo Fisher, MA, USA) to exclude protein-based interactions in the subsequent binding studies.

### 4.9. Immunoblotting

*L. pneumophila* JR32 was lysed by sonication (Bandelin, Sonopuls, Berlin, Germany) in a lysis buffer (150 mM sodium chloride, 1% Triton X-100, 50 mM Tris; pH 8.0), followed by Proteinase K treatment (0.05 mg/mL). Afterwards, the samples were pipetted onto a nitrocellulose membrane (Macherey-Nagel, Düren, Germany), and an immunoblot test was performed as described by Raulf et al. [46]. Briefly, the membrane was blocked with a 5% milk solution and then incubated with the respective CLR-Fc fusion proteins (1 µg/mL). For detection, an HRP-conjugated goat anti-human IgG antibody (Jackson ImmunoResearch Labs, RRID: AB_2337586) was diluted 1:10.000 in TBS + 0.05% Tween-20. The signals were visualized using Amersham ECL Prime Western Blotting Detection Reagent (Thermo Fisher, MA, USA). Chemiluminescence was measured using a chemiluminescence imager (Bio-Rad Laboratories, Hercules, CA, USA).

### 4.10. Isolation and Differentiation of Murine BMMs

Bone marrow cells from WT and *Clec12a*^−/−^ mice were isolated from the tibia and femur of 8- to 16-week-old mice of both sexes. The bones were washed in 70% ethanol and flushed with IMDM + 10% FCS + 2mM L-glutamine + 100 U/mL pen/strep. The collected cell suspension was passed through a 40 µm cell strainer and centrifuged at 300× *g* for 5 min. RBC lysis (90% 160 mM NH_4_CL + 10% 100mM Tris-HCL (pH 7.5)) was performed, and bone marrow cells were washed and stored at -150 °C in 10% DMSO. For differentiation into bone-marrow-derived macrophages (BMMs), cells were cultivated in a BMM growth medium (RPMI 1640 + 20% FCS + 30% L929 fibroblast supernatant + 4.5 mM L-glutamine + 100 µg/mL pen/strep) for 10 days, and fresh medium was added on Day 4 of cultivation. Differentiated cells were replated in a BMM replating medium (RPMI 1640 + 10% FCS + 15% L929 fibroblast supernatant + 4.5 mM L-glutamine) at a density of 4 × 10^5^ cells/mL in 48-well plates one day before the experiment.

### 4.11. Isolation of Human Alveolar Macrophages

Primary human alveolar macrophages (AM) were isolated by repeated perfusion of the human lung tissue (with ethical approval from Charité EA2/079/13) with HBSS as described previously [47].

### 4.12. Human BLaER1 Cell Transdifferentiation and Generation of a Human BLaER1

#### CLEC12A^−/−^ Line

Human BLaER1 B-cells were kindly provided by Prof. Thomas Graf (Center for Genomic Regulation, Barcelona, Spain). The cells were grown in a BLaER1 cultivation medium (RPMI 1640 + 10% FCS + 100 µg/mL pen/strep) and passaged every 4 days. For transdifferentiation into BLaER1-derived macrophages, the cells were seeded into a 48-well plate in a BLaER1 differentiation medium (RPMI 1640 + 10% FCS + 100 µg/mL pen/strep + 10 ng/mL IL-3 and 10 ng/mL M-CSF + 100 nM β-estradiol). The cells were incubated for 6 days, and the stage of differentiation was confirmed via flow cytometry by evaluating the expression of CD11b and CD19. The generation of a BLaER1 *CLEC12A*^−/−^ cell line was achieved using the CRISPR/Cas9 system. The CAS9 enzyme and specific guide RNA (GCTGGACGCCATACATGAGA) (IDT, IA, USA) were assembled in vitro, and the ribonucleoprotein was mixed with the cell suspension, followed by electroporation in a 4-D nucleofector (Lonza, Basel, Switzerland) with the program EH-140.

Electroporated cells were collected in a prewarmed medium and incubated at 37 °C and 5% CO_2_ for 72 h. Single cells were sorted by flow cytometry and expanded in a 96-well plate and incubated at 37 °C and 5% CO_2_. Screening for indel mutations was achieved by extracting the DNA from 10^5^ cells of each clone and performing PCR, followed by Sanger sequencing. The primers used for PCR and sequencing were as follows: *CLEC12A* forward primer: tgacatgccacaattgtctactca; reverse primer: ttgccaagactcccaatccaa. Loss-of-function mutations in the BLaER1 clones were confirmed by flow cytometry.

### 4.13. Short-Term Infection of Murine BMMs and Human BLaER1 Cells

Mouse BMMs and human BLaER1 cells were seeded in 48-well plates and infected with *L. pneumophila* JR32 WT and *ΔflaA* at MOI 10. Infected cells were centrifuged at 200× *g* for 5 min and then incubated for 8 h (gene expression) or 18 h (cytokine production) at 37 °C and 5% CO_2_, respectively.

### 4.14. Quantitative RT-PCR

Total RNA was isolated from the cultured cells or lung homogenates using the RNeasy Plus Mini purification system (QIAGEN, Düsseldorf, Germany) or Trizol (Life Technologies, Darmstadt, Germany), respectively. Total RNA was reverse-transcribed using the high-capacity reverse transcription kit (Applied Biosystems, Darmstadt, Germany), and quantitative PCR was performed using TaqMan assays for murine *Ifnb1* and human *IFNB1* as well as human *IL1B* and *TNFA* (Thermo Fisher, MA, USA), on a qTOWER^3^ G instrument (Analytik Jena AG, Jena, Germany). The input was normalized to the average expression of murine and human GAPDH, and the relative expression (relative quantity, RQ) of the respective gene in untreated cells or PBS-treated mice was set to 1.

### 4.15. ELISA

The cytokine levels of TNFα, IFNβ and IL-6 from homogenized lung lysates of mice were measured with the LEGENDplex Mouse Anti-Virus Response Panel Multi-Analyte Flow Assay Kit (BioLegend, CA, USA). The concentrations of human IP-10 as well as human IL-1β and human TNFα (R&D System, MN, USA) and murine TNFα (Thermo Fischer, MA, USA) in the supernatants from cells infected in vitro were quantified by commercially available sandwich ELISA kits. The protein concentrations were determined in a FilterMax F5 Multi-Mode Microplate Reader (Molecular Devices, Sunnyvale, CA, USA) at 450 nm.

### 4.16. Flow Cytometry Assay

For determination of the number of specific cell populations in murine lungs, the cell suspensions were labeled with anti-CD45.2 (BioLegend, CA, USA), anti-CD11b (BioLegend, CA, USA), anti-Ly6G (BioLegend, CA, USA), anti-Ly6C (BioLegend, CA, USA) and anti-CD11c (BioLegend CA, USA). Additionally, cells were labeled with a live–dead fixable viability cell marker (Thermo Fischer, MA, USA). The differentiation stages of the BLaER1 cells were assessed by labeling the cells with anti-CD19 (BioLegend, CA, USA) and anti-CD11b (BioLegend, CA, USA). Confirmation of functional CLEC12A knock-out was evaluated by labeling with anti-CLEC12A (Milteny Biotec, Bergisch Gladbach, Germany). The analyses were conducted on a Becton Dickinson LSRFortessa flow cytometer using FACS DIVA software (BD Bioscience). Data analysis was carried out using FlowJo software (Tree Star).

### 4.17. Intracellular Bacterial Replication Assay

Murine BMMs, human BLaER1 cells and human alveolar macrophages were infected with a MOI of 0.1. Plates with the bacterial suspension added to the cells were centrifuged at 200× *g* for 5 min and subsequentially incubated at 37 °C and 5% CO_2_. After 30 min, the cells were washed with PBS and cell media supplemented with 50 µg/mL gentamycin. After 1 h, cells were washed twice with PBS, and fresh medium was added. The cells were incubated at 37 °C and 5% CO_2_, and lysis was performed with a 1% saponin solution at 2, 24, 48 and 72 h after infection. The combined CFUs of the cell lysate and the supernatants were evaluated by plating a defined volume of different serial dilutions on BCYE agar.

### 4.18. Statistical Analyses

All data are presented as the mean ± SD. The data were analyzed using GraphPad Prism Version 8 (GraphPad Software, La Jolla, CA, USA), and paired Student’s t-tests or multiple paired t-tests were performed for the in vitro assays, while the in vivo assays were analyzed using a two-way ANOVA and the Mann–Whitney U-test. Asterisks indicate significant differences (n.s. = not significant, * *p* < 0.05, ** *p* < 0.01, *** *p* < 0.001).

## Figures and Tables

**Figure 1 ijms-24-03891-f001:**
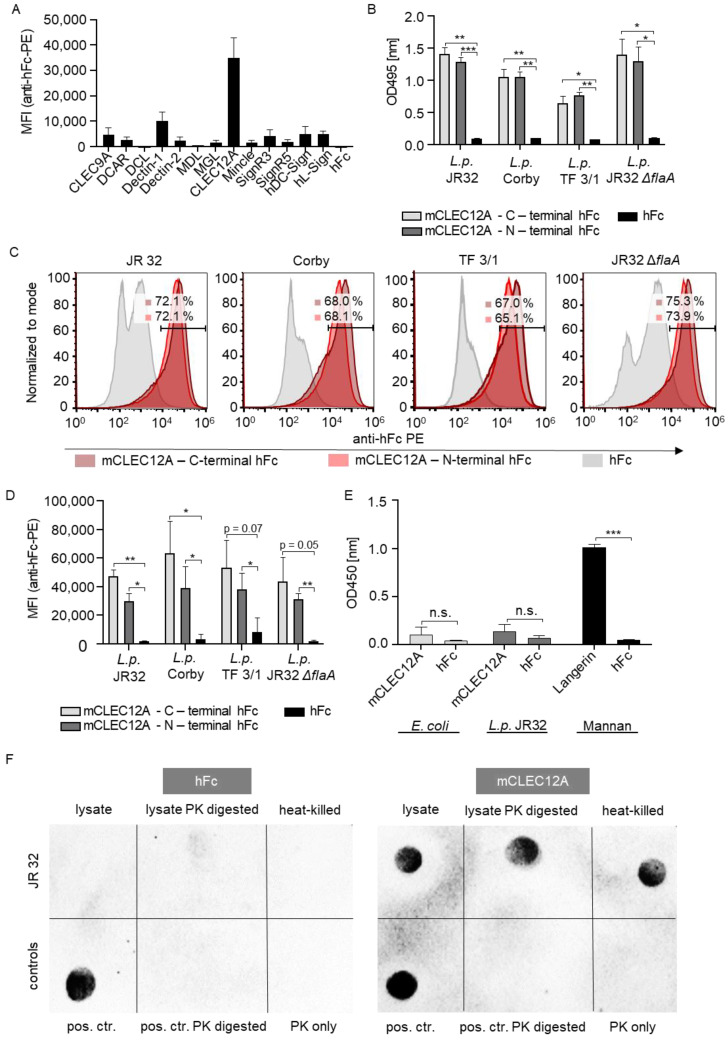
The murine CLEC12A-hFc fusion protein recognizes *L. pneumophila* (**A**) Flow cytometry-based binding study of a comprehensive CLR-Fc fusion protein library to *L. pneumophila* wild-type strain JR32. A PE-conjugated anti-hFc antibody was used for CLR detection. The results are presented as the mean fluorescence intensities (MFI). (**B**) ELISA-based binding study using the *L. pneumophila* strains JR32, Corby, TF 3/1 and JR32 *ΔflaA*. C-terminal hFc (N-CTLD-hFc-C) and N-terminal hFc (N-hFc-CTLD-C) murine CLEC12A fusion proteins were incubated with the respective strains, and binding was detected using an anti-hFc HRP antibody. (**C**,**D**) Flow cytometry-based binding study using the *L. pneumophila* strains JR32, Corby, TF 3/1 and JR32 *ΔflaA*. The binding study was performed with C-terminal hFc (N-CTLD-hFc-C) and N-terminal hFc (N-hFc-CTLD-C) murine CLEC12A fusion proteins. The detection was performed using a PE-conjugated anti-hFc antibody. (**C**) Representative histograms of flow cytometry-based binding studies are shown. Values within the histograms show the percentage of *L. pneumophila* strains that were positive for a PE-conjugated anti-hFc antibody. The first values represent the percentages of the C-terminal hFc murine CLEC12A fusion proteins binding to the respective *L. pneumophila* strain; the second values represent the percentages of the N-terminal hFc murine CLEC12A fusion protein. (**D**) The results of the flow cytometry-based binding studies are presented as the mean fluorescence intensities (MFI). (**E**) ELISA-based binding study using LPS isolated from *E. coli* and *L. pneumophila* JR32. The binding of the CLR Langerin to its ligand (mannan) was included as a positive control. (**F**) Representative dot blot (of 3 independent experiments) using *L. pneumophila* JR32. The *L. pneumophila* JR32 samples were pipetted on the upper row as bacterial lysate (“lysate”), proteinase K-digested lysate (“lysate PK digested”) and heat-killed untreated *L. pneumophila* (“heat-killed”) (from left to right). On the bottom, the following controls were pipetted: hFc as a positive control (“pos. ctr.”), proteinase K-digested hFc (“pos. ctr. PK digested”) and proteinase K only (“PK only”) (from left to right). The left membrane was incubated with hFc (“hFc”) as a control; the right membrane was incubated with murine CLEC12A—C-terminal hFc (“mCLEC12A”). The detection was performed using an anti-hFc HRP antibody. (**A**–**E**) All data are shown as the mean ± SD (n = 3). Data were analyzed using a paired Student’s t-test. Asterisks indicate significant differences (n.s. = not significant, * *p* < 0.05, ** *p* < 0.01, *** *p* < 0.001).

**Figure 2 ijms-24-03891-f002:**
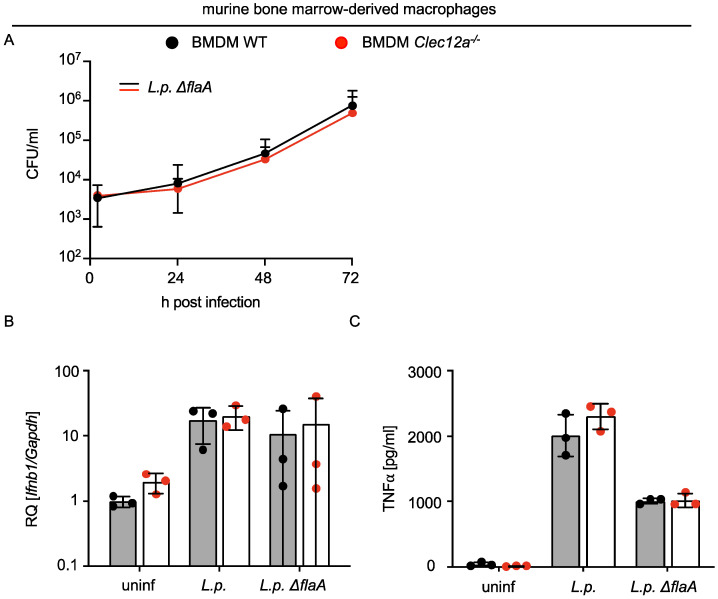
Replication of *Legionella* in murine BMDMs as well as type I IFN responses and TNFα production are not influenced by CLEC12A. (**A**) WT and *Clec12a*^−/−^ BMDMs were infected with *L. pneumophila ΔflaA* at MOI 0.1, and replication was assessed by evaluating CFUs in the cells and supernatants after 2, 24, 48 and 72 h. (**B**,**C**) WT and *Clec12a*^−/−^ BMDMs were infected with *L. pneumophila* JR32 (“*L.p*.”) or *ΔflaA* at MOI 10, and the expression of *Ifnb1* was evaluated at 8 h after infection by qRT-PCR. TNFα levels were quantified from supernatants after 18 h. All data represent the mean ± SD of 3 independent experiments carried out in triplicate. Differences were assessed using multiple paired *t*-tests. Comparisons with *p* < 0.05 were considered significant.

**Figure 3 ijms-24-03891-f003:**
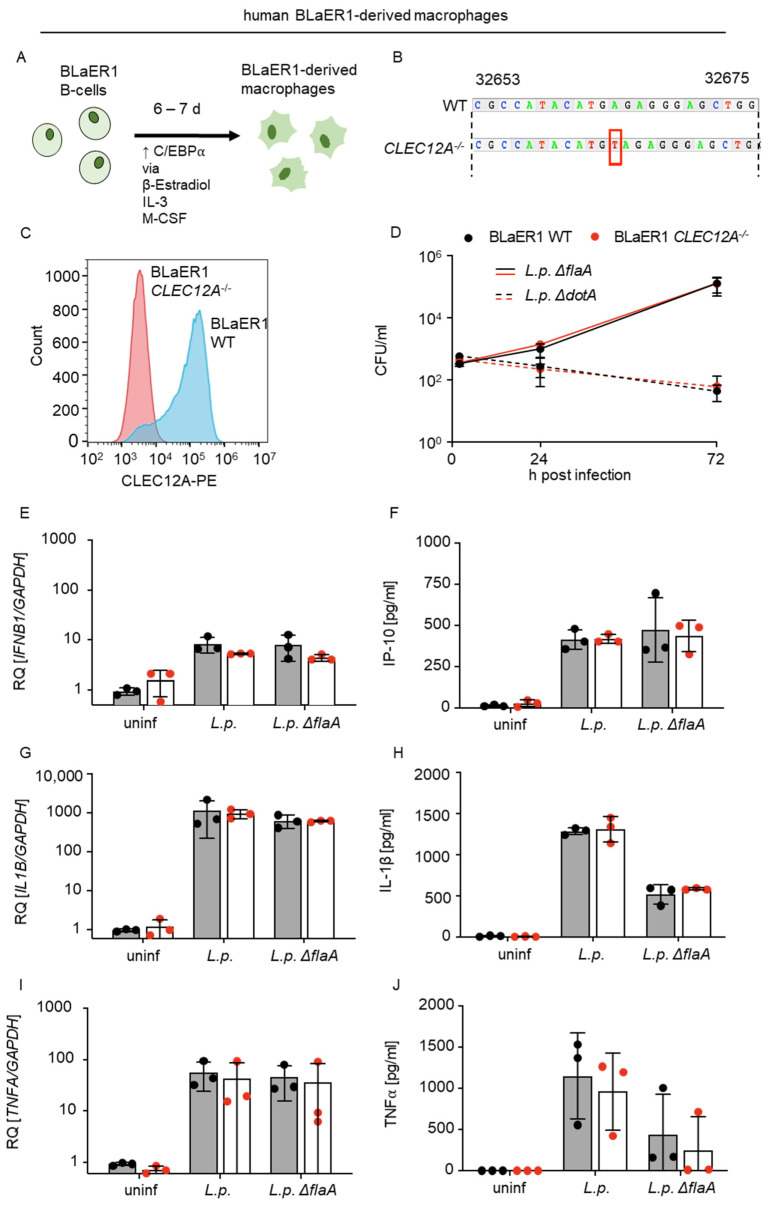
CLEC12A does not affect the replication of *L. pneumophila* or the production of proinflammatory cytokines and type I IFNs by BLaER1-derived human macrophages. (**A**) BLaER1 cells were transdifferentiated into BLaER1-derived macrophages by stimulation of the transcription factor C/EBPα with β-estradiol, IL-3 and M-CSF for 5 to 7 days. (**B**) BLaER1 *CLEC12A*^−/−^ cells were generated by introducing a frameshift of one base into CLEC12A (see red box) by CRISPR/Cas9. (**C**) The loss of CLEC12A in BLaER1 cells was confirmed by flow cytometry, using an anti-CLEC12A-PE labeled antibody. (**D**) The replication of *L. pneumophila* (*L.p.*) *ΔflaA* and *ΔdotA* in BLaER1 WT and *CLEC12A*^−/−^ cells was assessed by infecting cells at MOI 0.1 and evaluating CFUs at 2, 24 and 72 h after infection. BLaER1 WT and *CLEC12A*^−/−^ cells were infected with *L.p.* WT and *ΔflaA* at MOI 10. The expression levels of *IFNB1* (**E**), *IL1B* (**G**) and *TNFA* (**I**) were measured after 8 h by qRT-PCR and compared with uninfected controls. Data are shown as the relative quantification (RQ) of the target mRNAs relative to *GAPDH*. (**F**,**H**,**J**) Production of IP-10 (CXCL10) (**F**), IL-1β (**H**) and TNFα (**J**) was measured in supernatants of the infected BLaER1 WT and *CLEC12A*^−/−^ cells after 18 h. (**D**–**J**) All data represent the mean ± SD of 3 independent experiments carried out in triplicate. Differences were assessed using multiple paired *t*-tests. Comparisons with *p* < 0.05 were considered significant.

**Figure 4 ijms-24-03891-f004:**
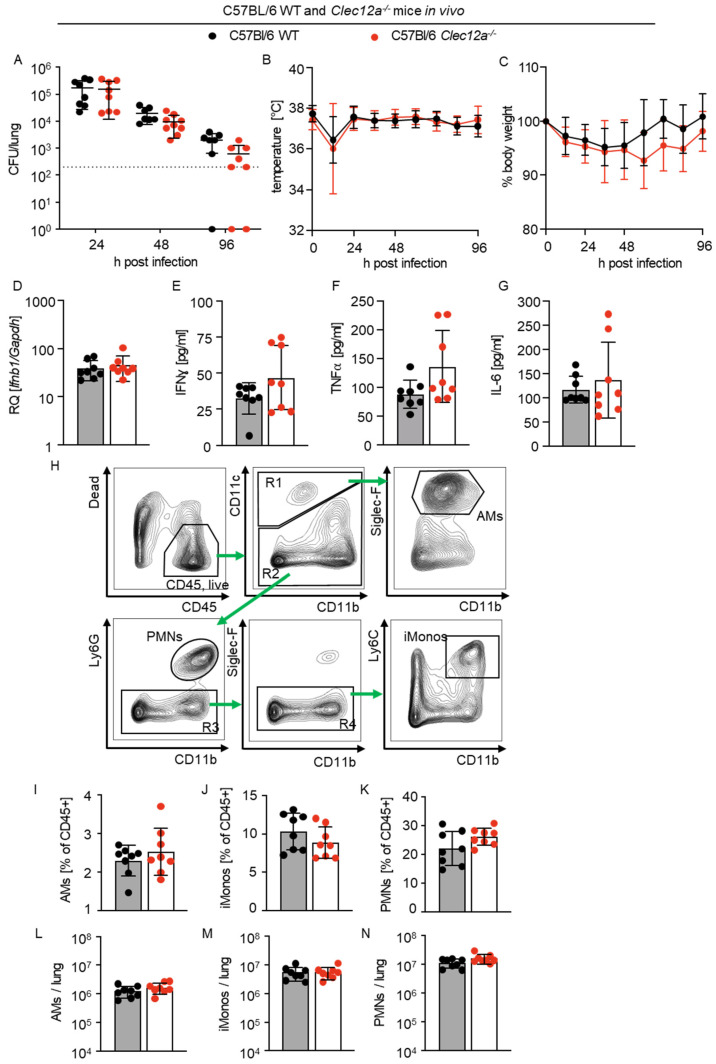
CLEC12A plays a limited role in lung infection by *L. pneumophila* in mice. (**A**) WT and *Clec12a*^−/−^ C57BL/6 mice were infected with *L. pneumophila* at a dose of 10^6^ CFU/mouse, and the bacterial loads from their lungs were assessed at 24, 48 and 96 h after infection by plating serial dilutions of homogenized lungs on BYCE agar. (**B**,**C**) Infected WT and *Clec12a*^−/−^ C57BL/6 mice were monitored for temperature (**B**) and body weight (**C**) over the course of the infection experiment. The expression levels of *Ifnb1* in homogenized murine lungs 24 h after infection were assessed by qRT-PCR and normalized to the lungs of PBS-treated WT mice (**D**). (**E**–**G**) The levels of IFNγ, TNFα and IL-6 in mouse lungs were measured 24 h after infection. (**H**) Representative flow cytometric analyses of alveolar macrophages (AMs; CD45^+^, CD11c^+^, CD11b^−^, Siglec-F^+^), inflammatory monocytes (iMonos; CD45^+^, CD11c^−^, CD11b^+^, Ly6C^+^) and polymorphonuclear neutrophils (PMNs; CD45^+^, CD11c^−^, CD11b^+^, Ly6G^+^). Green arrows indicate the gating strategy. (**I**–**N**) Percentages and numbers of AMs, iMonos and PMNs in the lungs of WT and *Clec12a*^−/−^ mice 24 h after infection. All data represent the means ± SD of 2 independent experiments, with 4 to 5 mice per experiment. Differences were assessed using a two-way ANOVA and the Mann–Whitney U-test. Comparisons with *p* < 0.05 were considered significant.

## Data Availability

All supporting information is provided in the submitted Appendix A.

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
