# Peer review of "CLEC12A Binds to Legionella pneumophila but Has No Impact on the Host’s Antibacterial Response"

_ijms, 2023, doi:10.3390/ijms24043891_

Round 1

Reviewer 1 Report

1- CLR detection by ELISA doesn't explain why first purified antigens were not used, especially when bacteria extracts are sometimes prone to false-positive results. 

 2- the paper did not explain why microscopy was not used to direct visualize CLR/bacteria interactions.

Reviewer 2 Report

In the manuscript entitled “CLEC12A binds to Legionella pneumophila but has no impact on the antibacterial host response”. The author investigated the role of CLEC12A in response to Legionella lung infection. They show that CLEC12A deficiency is unable to alter the antibacterial and inflammatory effects both in vitro and in vivo. Overall, the conclusion is well-supported by the results. The manuscript is very well written, and the experiments are well performed. I have several concerns and questions which I believe once resolved would strengthen the submission.

1. The study has limited rigor and reproducibility. Some experiments have a very small N and should be repeated to ensure that these findings are robust. For instance, Figure 3E-G.

2. More detailed information is needed in Materials and Methods. Where are these C57BL6/J WT and Clec12a-/- mice from? Did the author validate the Clec12a-/- mice using an antibody-based method?

3. The functional study is fully based on loss-of-function methods. Did the author test CLEC12A function after overexpression in macrophages?

Author Response

Pleaase see the attachment

Reviewer 3 Report

In this manuscript, the authors found the C-type lectin receptor CLEC12A can bind L. pneumophila, but CLEC12A deficiency did not affect the host antibacterial and inflammatory response. The experimental design, methodology, and analysis are excellent, even the results are negative.

Comments: 

  1.  CLEC12A does not affect L. pneumophila induced inflammatory cytokine TNFa and Ifnb1 levels, does it affect other cytokines? Such as IL-1b and IL-6? L. pneumophila induced Ifnb1 levels is low, so the authors used real-time PCR instead of ELISA.  
  2. It is interesting that CLEC12A can bind L. pneumophila,  even if the binding affects host response in vitro. So, it is important to find some clues about the CLEC12A function through in vivo study.  The authors did in vivo study that L. pneumophila infected WT and Clec12a-/- mice, neither cytokines nor innate inflammatory cells were significantly influenced by CLEC12A deficiency. Any difference in morbidity and mortality was found? Such as survival rate, body weight, lung injury (H&E staining)?  

Token together, the authors provide evidence that L. pneumophila interacts with host CLR, and this manuscript is useful to CLR studies although the function of binding is unknown. 
